# Is There Agreement and Precision between Heart Rate Variability, Ventilatory, and Lactate Thresholds in Healthy Adults?

**DOI:** 10.3390/ijerph192214676

**Published:** 2022-11-09

**Authors:** Letícia Nascimento Santos Neves, Victor Hugo Gasparini Neto, Igor Ziviani Araujo, Ricardo Augusto Barbieri, Richard Diego Leite, Luciana Carletti

**Affiliations:** 1Laboratory of Exercise Physiology (LAFEX), Physical Education and Sports Center, Federal University of Espírito Santo (CEFD-UFES), Vitória 29075-910, Brazil; 2Postgraduate Program in Physical Education and Sport, School of Physical Education and Sport of Ribeirão Preto, University of São Paulo (EEFERP-USP), São Paulo 05360-160, Brazil

**Keywords:** anaerobic threshold, lactate threshold, ventilatory threshold, exercise test, athletic performance, prescription

## Abstract

This study aims to analyze the agreement and precision between heart rate variability thresholds (HRVT1/2) with ventilatory and lactate thresholds 1 and 2 (VT1/2 and LT1/2) on a treadmill. Thirty-four male students were recruited. Day 1 consisted of conducting a health survey, anthropometrics, and Cardiopulmonary Exercise Test (CPx). On Day 2, after 48 h, a second incremental test was performed, the Cardiopulmonary Stepwise Exercise Test consisting of 3 min stages (CPxS), to determine VT1/2, LT1/2, and HRVT1/2. One-way repeated-measures ANOVA and effect size (η_p_^2^) were used, followed by Sidak’s post hoc. The Coefficient of Variation (CV) and Typical Error (TE) were applied to verify the precision. Bland Altman and the Intraclass Correlation Coefficient (ICC) were applied to confirm the agreement. HRVT1 showed different values compared to LT1 (lactate, RER, and R-R interval) and VT1 (V̇E, RER, V̇CO_2_, and HR). No differences were found in threshold 2 (T2) between LT2, VT2, and HRVT2. No difference was found in speed and V̇O_2_ for T1 and T2. The precision was low to T1 (CV > 12% and TE > 10%) and good to T2 (CV < 12% and TE < 10%). The agreement was good to fair in threshold 1 (VT1, LT1, HRVT1) and excellent to good in T2 (VT1, LT1, HRVT1). HRVT1 is not a valid method (low precision) when using this protocol to estimate LT1 and VT1. However, HRVT2 is a valid and noninvasive method that can estimate LT2 and VT2, showing good agreement and precision in healthy adults.

## 1. Introduction

Thresholds in physiology are essential concepts that establish the boundaries between different domains of exercise intensity (moderate/heavy/severe) [1,2]. In addition, thresholds can predict athletic performance, monitor training progress, and clinically assess a patient’s physiological function or dysfunction (e.g., in heart failure) [1]. Furthermore, some methods for threshold determination are costly, require qualified professionals, and need specific software. Thus, using more economical and less complex tools to determine thresholds is interesting. However, in this article, threshold 1 (T1) (e.g., first lactate, ventilatory or gas exchange, and heart rate variability (HRV) thresholds) lies between the moderate and heavy domains [2,3,4] and threshold 2 (T2) (e.g., second lactate threshold, second ventilatory threshold, or respiratory compensation point) lies between the heavy and severe domains [1,2,3,4,5]. Thresholds can be identified invasively by blood analysis (e.g., lactate, glucose, catecholamines) and noninvasively (e.g., gas exchange, electromyography, heart rate deflection point of heart rate, and infrared spectroscopy) [6,7,8]. Although different thresholds can be estimated, some still generate confusion about the moment of demarcation by indicating different points on the curve, so they still need to be studied and analyzed [1].

A potentially cheaper and less invasive method than LT and VT to identify thresholds can be performed through autonomic balance, determined by heart rate variability (HRV), which is defined by the oscillation in the interval between consecutive heartbeats as well as the oscillations between consecutive instantaneous heart rates [9].

The literature suggests using HRV to estimate physiological thresholds, such as the ventilatory or lactate threshold, in an incremental test. During the test, there was a progressive increase in exercise intensity over time with a concomitant reduction in HRV and parasympathetic activity, followed by vagal withdrawal [10,11]. The vagal withdrawal marks the first heart rate variability threshold (HRVT1), which occurs at approximately 50–60% of V̇O_2max_ [12] and may coincide with the first lactate and ventilatory thresholds (LT1 and VT1) [10]. In contrast, the second lactate threshold (LT2) and second ventilatory threshold (VT2) are also investigated to determine whether HRV can estimate them. However, little evidence has shown comparisons between VT2 and LT2 with the heart rate variability threshold (HRVT2), and even rarer are treadmill studies [13,14].

Ventilatory thresholds 1 and 2 [3,15] and lactate thresholds 1 and 2 [2,10] are validated and widely used methods for threshold identification. However, although they agree, their occurrence may not occur at identical work rates, i.e., with insufficient precision to be determined simultaneously [1,7]. Therefore, comparing both to a potentially cheaper and recent proposal (HRV) is essential.

As far as we know, no literature has shown the comparison between these six thresholds simultaneously (VT1/VT2, LT1/LT2, and HRVT1/HRVT2) on a treadmill. However, some authors compared three thresholds but only HRV1 with LV1 and LT1 on a cycle ergometer [10]. Other authors who used the first and second thresholds compared HRV only with the ventilatory thresholds (HRVT 1 vs. VT1 and HRVT2 vs. VT2) [13,14,16]. Therefore, it is interesting to present and compare the six thresholds in the same individual, allowing a more significant contribution to studies that demonstrate the thresholds in isolation, such as the comparison between only the first thresholds or only the second ones, and even only the comparison of HRV with the ventilatory threshold or with lactate threshold alone.

Furthermore, when LT, VT, and HRVT are involved, the studies used a cycle ergometer, which makes it difficult to transpose HRV behavior into a practical application such as running. Since V̇O_2_, ventilation, fiber recruitment patterns, and muscle coordination according to the specificity of each test or exercise can influence threshold identification [17], studies are needed to verify whether both thresholds (VT1/VT2 and LT1/LT2) can be estimated using HRVT1 and HRVT2 on a treadmill. Therefore, we aimed to test if there is agreement and precision between HRVT1/2 with LT1/2 and VT1/2 on a treadmill.

## 2. Materials and Methods

### 2.1. Participants

The study consisted of 34 healthy male university students who were physically independent (≥150 min·week^−1^ of physical exercise), 22 ± 2 years, height: 176.2 ± 6.6 cm, body mass: 72.89 ± 8.84 kg; body fat (%): 8.77 ± 3.69; BMI: 23.5 ± 2.66; and R-R intervals (ms): 935.7 ± 151.0). Informed consent was obtained from all subjects involved in the study. The study followed the ethical guidelines outlined in the Declaration of Helsinki and was approved by an ethical committee (CAAE: 76607717.5.0000.5542–18/09/2017). For this sample, the statistical power found was 80% (effect size f: 0.25-equivalent to the moderate η_p_^2^ found in the present study for (V̇O_2_); alpha: 5%), calculated by G*Power 3.1. University students aged 18 to 30 years were included, and none of the participants had cardiovascular, metabolic, respiratory, or osteoarticular diseases, and their maximal oxygen consumption (V̇O_2max_) was considered below 34 mL⋅kg^−1^⋅min^−1^ according to the American Heart Association [18]. Alcohol was forbidden for 48 h, and caffeine-containing products for 24 h prior to the beginning of the study.

### 2.2. Study Design

Data collection occurred over two days with a 48-h interval in the morning. On the first day, questionnaires to identify health status were applied, followed by anthropometry, an electrocardiogram, and a cardiopulmonary exercise test (CPx) applied by a cardiologist to familiarize and determine the initial intensity of the second day. A Cardiopulmonary Stepwise Exercise Test (CPxS) was performed on the second day to compare the six thresholds (LT1/2, VT1/2, HRVT1/2). Two evaluators analyzed the criteria of VTs, LTs, and HRVTs blindly and independently. When necessary, a third evaluator was requested. The intraclass correlation coefficient between evaluators was used with values varying between LT1: 0.889, LT2: 0.948; VT1: 0.965, VT2: 0.954; HRVT1: 1.00, and HRVT2: 1.00. HRV, V̇O_2_, and Lactate were collected during the CPxS every 3 min and after the test (10 min recovery). All participants were instructed to have a light meal at least 2 h before the test and avoid caffeinated and strenuous exercise 24 h before the tests.

### 2.3. Anthropometry

Body mass and height were measured using a digital anthropometric scale (Marte Scientific, L200, SP), and the Body Mass Index (BMI) was calculated. In addition, seven skinfolds were measured (subscapular, triceps, mid-axillary, pectoral, abdominal, suprailiac, mid-thigh) (Mitutoyo Cescorf, Porto Alegre, Brazil) to calculate the body fat percentage (%BF). To calculate %BF, it is necessary to calculate body density (BD), and Jackson and Pollock’s seven skinfold equation was used: BD = 1.11200000 − 0.00043499 (X_1_) + 0.00000055 (X_1_)^2^ – 0.00028826 (X_4_), where X_1_ = sum of seven skinfolds and X_4_ = age in Years [19]. The %BF for the anthropometric technique was estimated using the SIRI (1961) equation %BF = (495/BD) – 450 [20].

### 2.4. Cardiopulmonary Exercise Test (CPx)

Before the CPx, the individuals remained in the supine position on a stretcher for 5 min, where a 12-lead resting electrocardiogram was performed (USB Micromed electrocardiograph, Brasília, Brazil). In addition, electrocardiographic records were performed during pre-exertion, standing on the treadmill, and during exertion. Then, the CPx was performed on a treadmill (Inbra Sports Super ATL, Porto Alegre, Brazil), with a slope of 1% in the ramp protocol, with a duration of 8 to 12 min. The initial speed was 5 km·h^−1^ with gradual increments of 1 km⋅h^−1^ each minute, supervised by a cardiologist (room temperature: 23 °C–25 °C). At least four criteria were considered to define the CPx as the maximum test: (a) Voluntary exhaustion; (b) ≥ 90% of the age-adjusted estimate of HRmax; (c) z respiratory exchange ratio (RER) equal to or greater than 1.05; (d) V̇O_2max_ plateau or peak with increased exercise intensity; (e) peak blood [lactate] of ≥ 8 mM. During the CPx, a Metabolic Gas Analyzer (Cortex Metalyzer 3B, Leipzig, Germany) was used, with breath-by-breath measurement. Before each test, a gas mixture was used to calibrate (11.97% O_2_ and 4.95% CO_2_) the ambient gas. Then, the volume was calibrated with a 3 L syringe. Ventilation (V̇E), oxygen consumption (V̇O_2_), carbon dioxide production (V̇CO_2_), respiratory exchange ratio (RER), and speed were analyzed for an average of 15 s.

### 2.5. Cardiopulmonary Stepwise Exercise Test (CPxS)

The CPxS was used to identify the six thresholds (VT1/2, LT1/2, and HRVT1/2) for comparing and validating the HRVTs. CPxS was applied 48 h after CPx, following the same precautions and procedures described in CPx. The CPx determined the initial intensity of this protocol on the first day. The CPxS incremental test starts at 4 km⋅h^−1^ below the speed in VT1 determined by CPx with an increment of 1 km⋅h^−1^ every 3 min (considering the first two stages as a warm-up, with a slope of 1% until maximum effort), measuring lactate, gas analysis, and HRV. Before the test, the participants remained at rest for 10 min in the supine position to measure the HRV.

### 2.6. Blood Lactate Concentrations

Samples of 25 μL of capillary blood from the ear lobe were collected, at rest, during the CPxS test (every 3 min, participants jump off the treadmill and come back quickly in approximately 15 to 20 s, with no time wasted at each stage), and in passive recovery after 1, 3, 5, 7, and 10 min. The lactate peak in recovery was considered the highest value measured in this interval. The samples were stored under refrigeration (−4 °C) for further analysis on the equipment YSI 2300 STAT plus (Ohio, USA) [21].

### 2.7. Heart Rate Variability (HRV)

A Polar H7 Bluetooth heart rate monitor (Polar Electro Oy, Kempele, Finland) was connected to a smartphone to collect HRV, beat by beat (R-R intervals). The Polar H7 was validated to measure HRV data [22]. Each step described in CPxS was recorded in the Elite HRV app (HRV Elite, Asheville-North Carolina) [23] and edited in the Kubios HRV 3.0 Program. Data were collected at rest and during and after CPxS.

### 2.8. Determination of the Ventilatory Threshold (VT)

VT1 was identified by the increase in the ventilatory equivalent of O_2_ (V̇E/V̇O_2_) without increasing the ventilatory equivalent of CO_2_ (V̇E/V̇CO_2_) (Figure 1). When necessary, V-slope and CO_2_ excess were observed to help identification (Metasoft^TM^ software) [3,15]. VT2 was identified as the moment of the lowest point of V̇E/V̇CO_2_ with subsequent elevation, in addition to the moment of the gradual decrease in PetCO_2_ [3]. The same two evaluators always determined the VT1 and VT2, and a third opinion was requested in case of disagreement. The participants were excluded when it was not possible to identify the VT (a success rate of 85.3% for T1 and 100% for T2). All intervals between steps, such as when participants jumped off the treadmill to take blood samples, were excluded from the analysis to avoid misinterpretation. The average for the last 30 s of each stage was analyzed.

### 2.9. Determination of the Lactate Threshold (LT)

The LT was identified by the visual method. LT1 was determined by the first increase in the blood lactate concentration above resting values [2,10] and LT2 by the second linearity breakpoint and exponential lactate accumulation [2,5,24] (Figure 1).

### 2.10. Determination of the Heart Rate Variability Threshold (HRVT)

The R-R intervals were grouped into three-minute sequences for HRV analysis. Data filtering was performed using the Kubios HRV Standard 3.0 Program (Gronau, Germany), and it was automatically filtered to remove missing or premature intervals and artifacts not exceeding 5% [10]. The first 90 s of physical effort at each stage were excluded from the analysis due to the HR and HRV kinetic adjustment. This study chose the Root Mean Square of the Successive Differences (RMSSD) and Poincaré plot indexes (SD1 and SD2) as the HRV indexes (time-domain and non-linear HRV parameters). HRVT1 was identified by the first linearity breakpoint determined by visual inspection for the RMSSD variables, and when necessary, SD1 was used to confirm the identification [25,26]. HRVT2 was determined by the linearity breakpoint after the lowest value with a subsequent increase in the RMSSD and confirmed by the linearity break of the SD1/SD2 variables only when necessary (determined by visual inspection) [26,27,28]. They were identified in Excel software (Microsoft Excel^®^ 2022) (Figure 1). Only two evaluators were necessary for HRVT identification because there was no disagreement. The same evaluators were used to determine VTs, LTs, and HRVTs, and they did not have experience with HRVTs. All thresholds (LT1/2, VT1/2, HRVT1/2) were represented in Figure 1 (individual example) in CPxS.

### 2.11. Statistical Analysis

All data were tabulated and double-verified by independent researchers. The analysis was performed using SPSS 20.0 software, and the figures were created by Excel software and GraphPad Prism 6. The normality was tested using the Shapiro–Wilk test and submitted to evaluate the histogram, kurtosis, and skewness. The results were presented as mean ± standard deviation (SD). Student’s *t*-test was used to compare the maximum values of CPx with CPxS. A repeated-measures ANOVA was used to compare the methods of identifying VT with LT and HRVT, followed by Sidak’s *post-hoc* (T1, n = 29 and T2, n = 34). The Greenhouse-Geisser correction was considered when a lack of sphericity was noted (F statistics, degrees of freedom, degrees of freedom error). Partial eta-squared (η_p_^2^) was utilized as a measure of effect size in ANOVA, using small (η_p_^2^ = 0.01), medium (η_p_^2^ = 0.06), and large (η_p_^2^ = 0.14) effects [29]. The typical error (TE = SD_diff_/$\sqrt{2}$; where SD_diff_ is the standard deviation of difference) was expressed as an absolute value and as a percentage of the mean value, and the coefficient of variation (CV) was expressed in a percentage [CV = (SD/$\bar{X}$)·100; where SD is the standard deviation of data and $\bar{X}$ is the mean], both used to test precision [30,31]. Bland Altman plots were used to identify agreement between the different methods (one-way ANOVA was used to compare the bias between the methods). The intraclass correlation coefficient (ICC) was used to scale the agreement (reliability). The reference values for the ICC were < 0.5 (poor), 0.5–0.75 (moderate), 0.75–0.90 (good), and ≥0.90 (excellent) [32]. Statistical significance was set at *p* < 0.05 for all analyses.

## 3. Results

The maximal values at CPxS are demonstrated in Table 1.

The average initial velocity of participants in the CPxS was 5.3 ± 1.2 km·h^−1^. HRVT1 showed statistically higher values when compared to LT1 for the variables lactate and RER (*p* < 0.05) and lower for the R-R interval in LT1 (Table 2). HRVT, V̇E, RER, V̇CO_2_, and HR were significantly higher than VT1 (*p* > 0.05). The R-R intervals were lower than VT1 and LT1 (*p* < 0.05) (Table 2). For the other variables, no statistical differences were found between LT1 and VT1 (*p* > 0.05) (Table 2). The percentage demonstrated in Table 2 and Table 3 (speed, V̇O_2_, RR, and HR) is relative to the maximum value reached during the CPxS.

No statistical differences were found in the methods’ comparison: HRVT2 vs. LT2, HRVT2 vs. VT2, and LT2 vs. VT2 (*p* > 0.05) (Table 3).

Furthermore, repeated-measures ANOVA revealed a significant effect mainly for the variables Lactate (F_2, 56_ = 4.83, *p* = 0.012, η_p_^2^ = 0.147, large), RER (F_2, 56_ = 18.06, *p* = 0.000, η_p_^2^ = 0.392, large), RR (F_2, 56_ = 6.56, *p* = 0.003, η_p_^2^ = 0.190, large), and HR (F_2, 56_ = 5,12, *p* = 0.009, η_p_^2^ = 0.155, large) between methods in T1 (Table 2), but no significant effects were found in T2 (Table 3).

At T1 (LT1, VT1, and HRVT1), speed, V̇O_2_, and HR demonstrated TE values greater than 10% (Table 4). At T2 (LT2, VT2, and HRVT2), all variables showed TE lower than 10% (Table 4). In T1, velocity, HR, and V̇O_2_ present CV > 12%, while T2 CV < 12% for the same variables, showing that T2 had less variation than T1. The ICC at T1 was moderate (*p* < 0.05), and for T2, the ICC was moderate to good (*p* < 0.05).

There were no statistical differences between the means of differences for all methods (*p* > 0.05) (Figure 2, Figure 3 and Figure 4).

## 4. Discussion

As far as we know, this is the first study to compare these six thresholds on a treadmill (LT1/LT2, VT1/VT2, and HRVT1/HRVT2). The main findings of our study suggest that HRVT2 has good agreement and precision to estimate LT2 and VT2 in healthy adults, increasing the ecological validity of these noninvasive methods and allowing a real-world HRVT1/2 determination. In contrast, HRVT1 did not show enough precision to estimate LT1 and VT1.

### 4.1. LTs vs. VTs

Lactate and ventilatory thresholds presented a moderate to good intraclass correlation and no statistical difference. However, it is essential to highlight that LT1 vs. VT1 demonstrated less precision (CV > 12% in all variables) and ICC (<0.750) than LT2 vs. VT2. Some studies also found different results when comparing these methods (LT1 vs. VT1 or LT2 vs. VT2) [8,10,14,16,25,33], which made our findings important to confirm the use of both LTs and VTs to estimate the exercise intensity. VT1 is challenging to identify due to several factors that can influence the agreement and precision identification, such as the cardiorespiratory fitness level [34], walking-running transition [35], control of food intake [1], and ergometer used [17]. Still, the present study showed an 85.3% success rate in identifying VT1, similar to research on a cycle ergometer [10], and a 100% success rate in second threshold identification. The CPx performed before CPxS likely affects the small loss percentage because it helps define the initial load of CPxS since the intensity of the protocol start can influence the identification of the VT1.

### 4.2. LT1 and VT1 vs. HRVT1

At HRVT1, Lactate, V̇E, RER, CO_2_, HR, and RR were statistically different compared to LT1 and VT1. However, no difference was found in speed, which could contribute to practical application in training (Table 2). However, more differences were found between HRVT1 and VT1 than HRVT1 and LT1 when observing the mean values on average (Table 2). HRVT1 seems closer to LT1 than VT1, especially for the HR parameter, which differed between HRVT1 and VT1, and it is among the three primary parameters for training prescription (speed, V̇O_2_, and HR). Still, we can infer that HRTV1 tends to overestimate the values of LT1 and VT1 for some parameters.

Most studies that compared HRVT1 with LT1 or VT1 demonstrated good agreement in V̇O_2_, HR, lactate, and speed [8,10,25,33]. However, one study showed difficulty in identifying VT1 using HRV analyses [16], which is close to our findings, reinforcing that T1 (HRVT1, VT1, LT1) is less precise to determine using different variables. On the other hand, some researchers who found good agreement at T1 only presented V̇O_2_ and did not present the parameter of speed or HR (km·h^−1^) [8,10], not allowing a comparative analysis with parameters such as speed (km·h^−1^). Our findings demonstrate a good mean difference (Bland Altman) with moderate ICC in V̇O_2_, velocity, and HR. However, we found low precision for all these variables (TE > 10% and CV > 12%) in T1 (LT1, VT1, HRVT1), which makes it difficult to show good acceptance of the HRVT1 to estimate LT1 and VT1.

Ramos-Campos et al. (2017) did not identify reproducibility between HRVT1 and VT1 for speed [14]. These results differed from the present study because no difference was found in speed in the present study. However, the precision was low. Besides that, both studies showed difficulties in determining VT1 using HRV analysis. Different methods to identify heart rate variability thresholds, such as the fixed value of 1 ms between stages, the visual method, and mathematical analysis, can make it difficult to compare different thresholds [8].

Despite that, the difficulty in determining VT1 is also essential because the occurrence or not of thresholds can result in a sample loss [10] and may harm the prescription of training intensity [36,37]. In addition, our study was careful to use an incremental treadmill test 48h before the Cardiopulmonary Stepwise Exercise Test to determine the initial intensity and minimize the loss of the first threshold. Different ergometers, treadmills, or cycle ergometers can be another problematic factor because they demonstrated different results. For example, in maximum progressive protocols on the cycle ergometer, many studies have accurately shown a relationship between vagal withdrawal and thresholds 1 (T1), determined by HRV, concentrations of blood lactate, and gas exchange [10,11,25,33]. Nevertheless, motor differences in exercise specificity in the different protocols are essential factors for determining thresholds and must be considered [17].

Consequently, protocols performed on a treadmill can present different behavior than the cycle ergometer, preferably at T1 [17]. Furthermore, the literature demonstrated that T1 could occur in the moment of transition from walking to running between 6 km·h^−1^ and 8 km·h^−1^ [35], which does not happen in the cycle ergometer. These findings are similar to our averages, in which they presented means of 7.15, 7.34, and 6.78 km·h^−1^ for LT1, HRVT1, and VT1, respectively.

Furthermore, the walk-to-run transition can modify some physiological variables, such as HR, V̇E, and V̇O_2_, causing interference in autonomic control, HRV responses, and gas exchange, creating confusion in threshold identification [35]. However, further research is needed to compare different ergometers using HRVT to determine T1 and confirm whether these transitions influenced the threshold identifications. Therefore, the use of HRVT1 to estimate LT1 and VT1 is not suggested under these conditions because, although they agree, their occurrence happened with insufficient precision. Another possibility to determine VT1 using HRV would be through the short-term scaling exponent alpha1 (DFA a1) [38]. However, it is still a recent proposal, in which the authors themselves suggest that the use of this index still needs to be better studied [38] since it appears to be influenced by spontaneous breathing during exercise [39], whereas this is less influenced when using RMSSD or the Poincaré plot [40].

### 4.3. LT2 and VT2 vs. HRVT2

HRVT2 demonstrated excellent agreement for all parameters compared to LT2 and VT2. Our study used the time domain and the Poincaré plot only when necessary to identify HRTV2 [26], which is simple to analyze and interpret since a third evaluator was not required. Furthermore, HRV measurement was performed with lactate and gas analysis, showing agreement and precision in using HRVT2 to estimate LT2 and VT2, which is helpful for monitoring and prescribing the intensity zones in exercise. Thus, research that evaluated VT2 and LT2 with HRVT2 demonstrated good agreement between the methods even in different sports modalities [14,41,42]. Some authors used variables in the frequency domain to identify T2 [42]. However, using these variables requires a more complex and careful analysis because the frequency domain is easily influenced by breathing, which may have less influence when applied to respiratory sinus arrhythmia (RSA) but is used only at rest and needs to have a fixed breathing rate [40]. However, changes in the breathing rate do not markedly influence the RMSSD or Poincaré plot dimension [40], which does not affect our results. Therefore, it is essential to use simple methods to identify HRVT2, such as the time domain [9].

These tests to estimate LT1/LT2 and VT1/VT2 by HRVT1/HRVT2 provide new insights into the relationship between the first and second thresholds. Some studies emphasize only the excellent agreement of the first thresholds, leaving the second thresholds out due to their wide range of identifications, and often do not show the exact timing [1,2]. However, this is not what our study demonstrated. Instead, we demonstrated excellent agreement on T2 (LT2 and VT2) estimated by HRV.

### 4.4. Importance to Use Visual Methods to Identify HRVTs

The methodological choices were constrained primarily by visual analysis to determine thresholds. It is easier to analyze (agreement in 100% of identifications) and identifies thresholds in the daily routine of health professionals because the use of the visual method allows identification and does not need expensive software (just need the free Kubios software or a Microsoft Office or Libre Office to analyze). Researchers strongly recommend determining T1 for health participants but are not highly trained, employing visual, mathematical, or a combination of both methods, as was performed in the present study [1,10]. Furthermore, some authors found that the visual method to determine HRV was validated in the literature and had better reliability than a mathematical method (Dmax) [27]. Therefore, all the methods used in this study are valid and recommended for all these reasons.

### 4.5. Importance of T1 and T2 in Training

Thresholds 1 and 2 are significant indices to help establish rigorous exercise intensity domains to individualize training and rehabilitation. They can help predict athletic performance, monitor training progress, and clinically assess the patient’s physiological function or dysfunction [1]. Exercise intensity is an important training element for effective cardiovascular and metabolic adaptations [43], mainly in T2. However, some researchers have also raised the importance of training volume even at low intensity (below or at T1), suggesting that duration is crucial to inducing training effects [44]. In this sense, researchers have demonstrated a polarized model of intensity distribution for training, in which participants who spend approximately 75% of the training below T1, 5–10% between T1 and T2, and 15–20% above T2 may have less autonomic stress, better motivation, and performance [36,37]. Consequently, the present study demonstrates the importance of determining the intensity in T1 and T2 in favor of better agreement and precision (accuracy) in practical and prescribing training. Future studies are needed to find better indices or methods to estimate T1. T1 is an important index when it comes to specific populations, for example, in patients with cardiovascular disease (ischemia or heart failure), who require adequate workload control of low-intensity exercises (below T1) [45]. In addition, the position statement of the European Association for Cardiovascular Prevention and Rehabilitation suggests the use of threshold-based exercise intensity prescriptions in heart disease patients [46,47].

### 4.6. Practical Significance

These findings may be helpful to coaches, conditioning instructors, and laboratories’ daily routines to monitor and prescribe exercise intensity. The HRV presented a simple analysis and interpretation method, visually using the time domain and Poincaré plot. Furthermore, HRV measurement using the heart rate sensor via a smartphone reduces the cost and facilitates its analysis, allowing less complexity to assess and help identify different intensity domains, which is essential for monitoring exercise intensity and prescriptions.

### 4.7. Limitations

Although the present study provides meaningful information, some limitations should be acknowledged. This study is cross-sectional, so future research should be considered to assess the training effect on HRVTs. In addition, studies are needed to measure HRV during exercise in different individuals with a wider range of BMI, ages, and fitness levels. The present study did not strictly control food intake or fluid intake in the pre-test, but all participants were instructed to eat 2 h before the test, and the same hydration was offered on the day of the test. Besides, all parameters were measured simultaneously and with the same conditions, impacting all thresholds equally. Furthermore, to obtain stable RR intervals, the 3-min stage protocol was used in this study; however, some researchers recommend the 1-min stage for better VT detection [10].

## 5. Conclusions

HRVT1 presents low precision using this protocol to estimate LT1 and VT1. However, HRVT2 is a valid and noninvasive method that can estimate LT2 and VT2, showing good agreement and precision in healthy adults. Therefore, studying HRVTs using this simple and visual method on a treadmill must be encouraged to show consistency and increase ecological validity.

## Figures and Tables

**Figure 1 ijerph-19-14676-f001:**
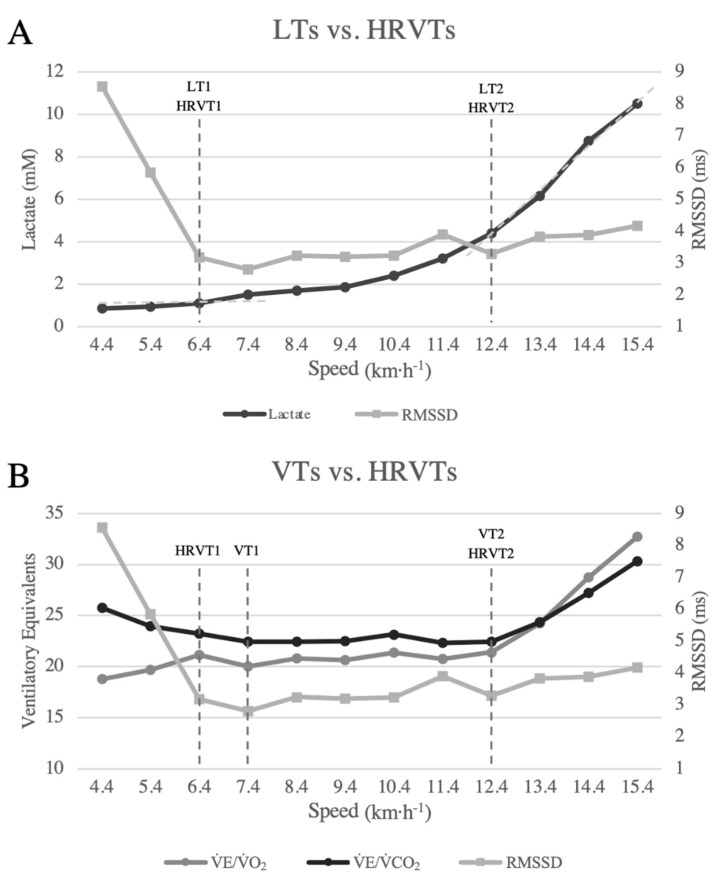
Represents an example of threshold identifications 1 and 2 on the CPxS (single participant). (**A**) represents the lactate threshold 1 and 2 (black line) and heart rate variability thresholds 1 and 2 (gray line). (**B**) represents the ventilatory threshold 1 and 2 (dark gray and black lines) and heart rate variability thresholds 1 and 2 (light gray line). Lactate thresholds 1 and 2 (LT1 and LT2), ventilatory thresholds 1 and 2 (VT1 and VT2), and heart rate variability thresholds 1 and 2 (HRVT1 and HRVT2).

**Figure 2 ijerph-19-14676-f002:**
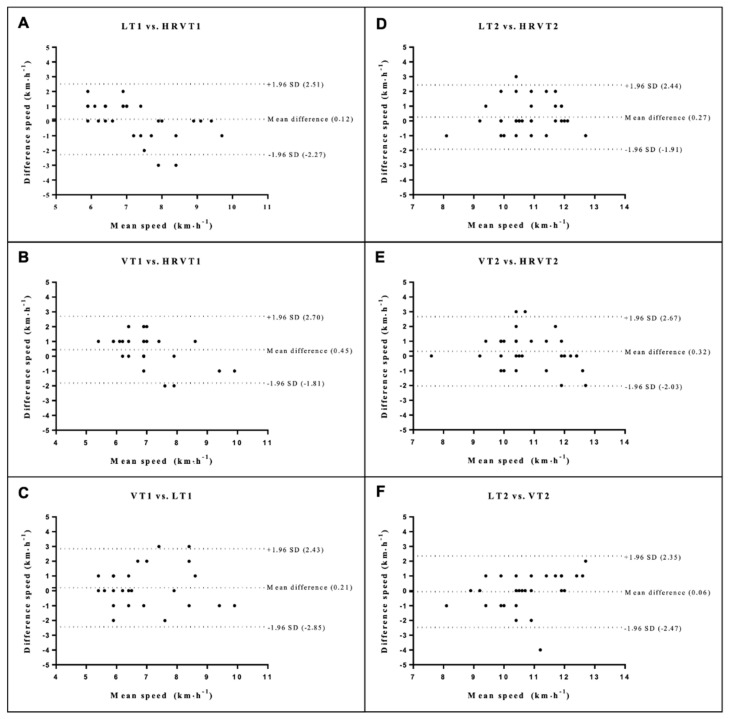
Bland Altman plots for speed at thresholds 1 (**A**–**C**) and 2 (**D**–**F**). The dashed lines at the ends represent the limits of agreement (1.96 SD) and the dashed central lines (bias). Lactate threshold 1 (LT1), ventilatory threshold 1 (VT1), heart rate variability threshold 1 (HRVT1), lactate threshold 2 (LT2), ventilatory threshold 2 (VT2), heart rate variability threshold 2 (HRVT2).

**Figure 3 ijerph-19-14676-f003:**
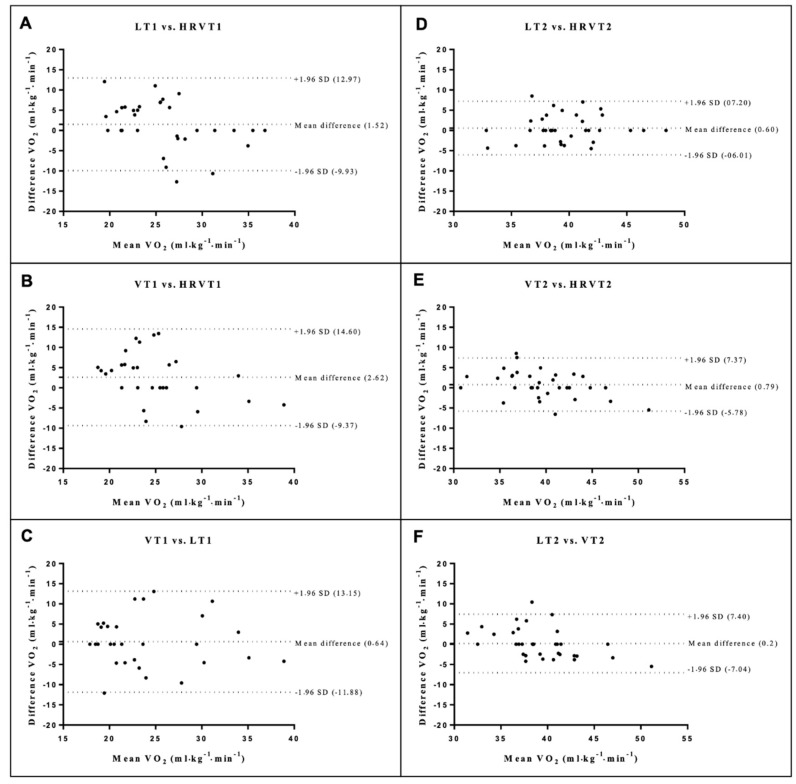
Bland Altman plots for V̇O_2_ at thresholds 1 (**A**–**C**) and 2 (**D**–**F**). The dashed lines at the ends represent the limits of agreement (1.96 SD) and the dashed central lines (bias). Lactate threshold 1 (LT1), ventilatory threshold 1 (VT1), heart rate variability threshold 1 (HRVT1), lactate threshold 2 (LT2), ventilatory threshold 2 (VT2), heart rate variability threshold 2 (HRVT2).

**Figure 4 ijerph-19-14676-f004:**
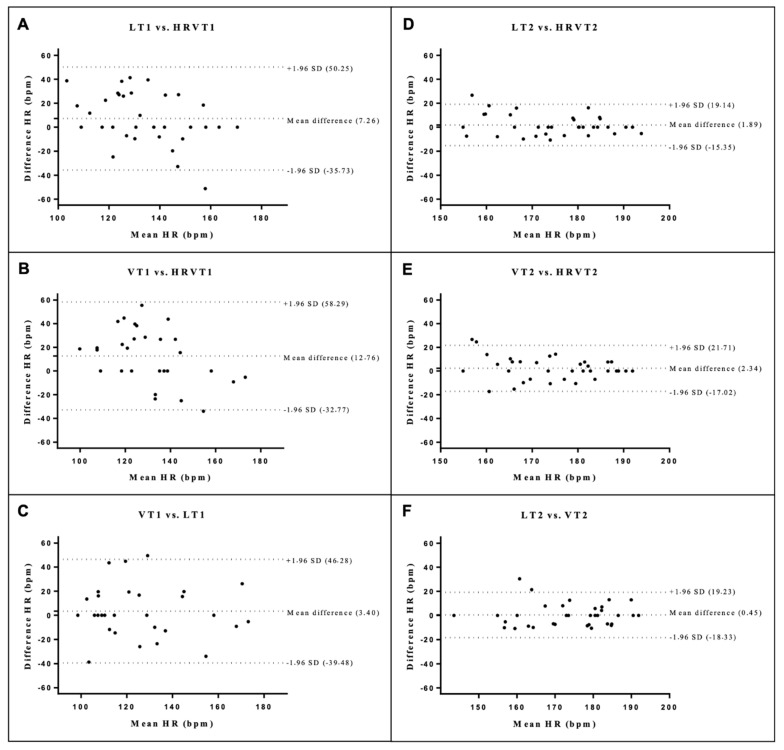
Bland Altman plots for HR at thresholds 1 (**A**–**C**) and 2 (**D**–**F**). The dashed lines at the ends represent the limits of agreement (1.96 SD) and the dashed central lines (bias). Lactate threshold 1 (LT1), ventilatory threshold 1 (VT1), heart rate variability threshold 1 (HRVT1), lactate threshold 2 (LT2), ventilatory threshold 2 (VT2), heart rate variability threshold 2 (HRVT2).

**Table 1 ijerph-19-14676-t001:** Maximal physiological characteristics of participants at CPxS.

Variables	Mean ± SD
HR_max_ (bpm)	194 ± 8
HR_max_ predicted (%)	97.75 ± 4.07
V̇E_max_ (L·min^−1^)	125.42 ± 15.91
V̇O_2max_ (ml·kg^−1^·min^−1^)	47.86 ± 4.96
V̇O_2max_ (L·min^−1^)	3.47 ± 0.42
RER_max_	1.02 ± 0.08
[La]_peak_ (mM)	10.47 ± 2.01

Mean ± SD; Stepwise Progressive Test (CPxS), Ventilation (V̇E), Respiratory Exchange Ratio (RER), Lactate ([La]).

**Table 2 ijerph-19-14676-t002:** Comparison between the means of the methods for identifying thresholds 1 in CPxS.

	LT1	VT1	HRVT1	Within-Participants Effects		
	Mean ± SD	Mean ± SD	Mean ± SD	F (df, df)	*p*	η_p_^2^
Speed (km·h^−1^)	7.2 ± 1.5	6.8 ± 1.4	7.3 ± 1.0	1.93 (2, 56)	0.155	0.064
Speed (%)	50.63 ± 8.68	49.26 ± 9.08	52.52 ± 5.01			
Lactate (mM)	1.57 ± 0.89	1.58 ± 0.85	2.01 ± 0.82 *	4.83 (2, 56)	**0.012 ^§^**	0.147
V̇E (L)	44.44 ± 15.51	39.80 ± 12.57	47.96 ± 9.68 ^†^	3.95 (2, 56)	**0.025 ^§^**	0.124
V̇O_2_ (ml·kg^−1^·min^−1^)	25.02 ± 6.48	23.75 ± 6.58	27.01 ± 5.22	2.94 (2, 56)	0.061	0.095
V̇O_2_ (%)	51.62 ± 13.43	50.28 ± 13.71	55.68 ± 8.43			
RER	0.84 ± 0.08	0.82 ± 0.07	0.88 ± 0.07 *^†^	18.06 (2, 56)	**0.000 ^§^**	0.392
V̇CO_2_ (L·min^−1^)	1.56 ± 0.53	1.43 ± 0.48	1.71 ± 0.34 ^†^	4.15 (2, 56)	**0.021 ^§^**	0.129
V̇O_2_ (L·min^−1^)	1.84 ± 0.55	1.72 ± 0.48	1.95 ± 0.32	2.36 (2, 56)	0.103	0.078
RR (ms)	473.8 ± 85.9	498.0 ± 93.0	439.2 ± 52.9 *^†^	6.56 (2, 56)	**0.003 ^§^**	0.190
RR (%)	52.04 ± 10.37	53.30 ± 9.33	47.63 ± 7.16 *^†^			
HR (bpm)	131 ± 23	125 ± 25	139 ± 16 ^†^	5.12 (2, 56)	**0.009 ^§^**	0.155
HR (%)	66.20 ± 7.59	64.53 ± 12.29	71.12 ± 11.14 ^†^			

Lactate threshold 1 (LT1), ventilatory threshold 1 (VT1), and heart rate variability threshold 1 (HRVT1). Ventilation (V̇E), relative oxygen consumption (V̇O_2_), respiratory exchange ratio (RER), carbon dioxide consumption (V̇CO_2_), R-R interval (RR), heart rate (HR), variable of heart rate variability (RMSSD), F statistics and degrees of freedom (F(df, df)), partial eta-squared (η_p_^2^). * (*p* < 0.05) different from LT1; ^†^ (*p* < 0.05) different from VT1; **^§^** (*p* < 0.05) ANOVA effect.

**Table 3 ijerph-19-14676-t003:** Comparison between the means of the methods for identifying thresholds 2 in CPxS.

	LT2	VT2	HRVT2	Within-Participants Effects		
	Mean ± SD	Mean ± SD	Mean ± SD	F (df, df)	*p*	η_p_^2^
Speed (km·h^−1^)	10.7 ± 1.1	10.6 ± 1.4	10.9 ± 1.2	1.45 (2, 66)	0.242	0.042
Speed (%)	76.59 ± 5.42	76.07 ± 7.22	78.49 ± 6.23			
Lactate (mM)	3.96 ± 1.44	3.95 ± 1.19	4.30 ± 1.36	1.77 (2, 66)	0.178	0.051
V̇E (L)	79.25 ± 13.57	75.83 ± 12.86	80.18 ± 11.08	1.92 (2, 66)	0.155	0.055
V̇O_2_ (ml·kg^−1^·min^−1^)	39.39 ± 3.60	39.19 ± 5.16	39.98 ± 3.86	0.96 (2, 66)	0.389	0.028
V̇O_2_ (%)	82.63 ± 6.32	81.97 ± 7.55	83.85 ± 6.79			
RER	0.92 ± 0.07	0.91 ± 0.06	0.92 ± 0.06	0.79 (2, 66)	0.460	0.023
V̇CO_2_ (L·min^−1^)	2.62 ± 0.39	2.59 ± 0.39	2.66 ± 0.31	0.86 (2, 66)	0.429	0.025
V̇O_2_ (L·min^−1^)	2.86 ± 0.38	2.84 ± 0.39	2.90 ± 0.31	0.89 (2, 66)	0.417	0.026
RR (ms)	347.2 ± 27.1	348.0 ± 27.0	342.8 ± 21.6	1.26 (2, 66)	0.291	0.037
RR (%)	37.85 ± 5.49	37.91 ± 5.28	37.46 ± 5.81			
HR (bpm)	174 ± 13	173 ± 13	176 ± 11	1.18 (2, 66)	0.313	0.035
HR (%)	89.68 ± 4.14	89.49 ± 4.83	90.72 ± 3.71			

Lactate threshold 2 (LT2), ventilatory threshold 2 (VT2), and heart rate variability threshold 2 (HRVT2). Ventilation (V̇E), relative oxygen consumption (V̇O2), respiratory exchange ratio (RER), carbon dioxide consumption (V̇CO2), R-R interval (RR), heart rate (HR), variable of heart rate variability (RMSSD), F statistics and degrees of freedom (F(df, df), partial eta-squared (ηp2). There were no differences between methods (*p* > 0.05).

**Table 4 ijerph-19-14676-t004:** Typical Error, Coefficient of Variation, and Intraclass Correlation Coefficient between the methods for identifying thresholds 1 and 2.

	LT1 vs. VT1	LT1 vs. HRVT1	VT1 vs. HRVT1	LT2 vs. VT2	LT2 vs. HRVT2	VT2 vs. HRVT2
**Typical Error absolute (%)**						
Speed (km·h^−1^)	0.95 (13.8)	0.86 (11.9)	0.81 (12.3)	0.87 (8.1)	0.78 (7.4)	0.85 (8.0)
V̇O_2_ (ml·kg^−1^·min^−1^)	4.51 (18.5)	4.13 (16.4)	4.32 (18.5)	2.61 (6.6)	2.38 (6.0)	2.37 (6.1)
HR (bpm)	15 (12.2)	16 (12.2)	16 (14.1)	7 (3.9)	6 (3.6)	7 (4.1)
**Coefficient of Variation**						
Speed (km·h^−1^)	19.31	16.82	16.27	11.56	10.28	11.14
V̇O_2_ (ml·kg^−1^·min^−1^)	26.13	22.44	23.97	9.40	8.50	8.47
HR (bpm)	17.10	16.29	17.56	5.52	5.03	5.66
**Intraclass Correlation Coefficient**						
Speed (km·h^−1^)	0.693 *	0.671 *	0.637 *	0.684 *	0.663 *	0.709 *
V̇O_2_ (ml·kg^−1^·min^−1^)	0.684 *	0.616 *	0.616 *	0.797 *	0.744 *	0.840 *
HR (bpm)	0.748 *	0.559 *	0.504 *	0.842 *	0.840 *	0.783 *

Lactate threshold 1 (LT1), ventilatory threshold 1 (VT1), heart rate variability threshold 1 (HRVT1), lactate threshold 2 (LT2), ventilatory threshold 2 (VT2), heart rate variability threshold 2 (HRVT2). Relative oxygen consumption (V̇O_2_) and heart rate (HR). * *p* < 0.05.

## Data Availability

Not applicable.

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
