# Peer review of "Is There Agreement and Precision between Heart Rate Variability, Ventilatory, and Lactate Thresholds in Healthy Adults?"

_ijerph, 2022, doi:10.3390/ijerph192214676_

Round 1
Reviewer 1 Report
The reviewer congratulates the authors since the current paper leading with the agreement and precision of the heart rate variability, ventilatory and lactate thresholds in healthy adults, is an exciting and well-written article. Future studies investigating these relationships running on the ground or in other endurance tasks are warranted.
General Comments
- The reviewer missed in the discussion section reference to a recently proposed detection of T1 throughout the DFA analysis of RR (PMID: 33519504, PMID: 33071812)
Specific Comments
Abstract
- Line 12, please correct "34" to text.
- Please introduce CV and TE for T1 and T2 in the abstract
- Report the lack of differences in velocities for T1 and T2.
Introduction
- Lines 52-54, Some recent papers indicate that DFA analysis over RR interval is a useful tool to monitor LT1 and VT1. Why did the authors not implement that analysis for HRV1 (i.e., HR complexity analysis)
Materials and Methods
- Was the study approved by an ethical committee? Please indicate it in the text.
- Line 98-99, please report ICC between evaluators for each of the six variables separately.
- Was the velocity/power on the treadmill registered?
- Anthropometry, Lines 105-108, the formulae (and reference) employed to calculate the body fat should be reported.
- Lines 132-133, Please report the participants' initial velocity (mean and SD) employed.
- HRVTs determination, Lines 178-182, How was the linearity breakpoint determined by visual inspection or some mathematical function? Please clarify that point.
Results
-In table 2 and 3, please explain % variables (Speed, VO2, HR), the reviewer guesses the percentage is relative to the maximum value reached during the same test, but it should be stated to avoid confusion.
References
- Lines 471, please correct reference 7, the title is missing
Author Response
We would like to thank the reviewer for evaluating our manuscript. We have addressed all the reviewers' concerns.
Response to Reviewer 1 Comments
General Comments
Point 1) The reviewer missed in the discussion section reference to a recently proposed detection of T1 throughout the DFA analysis of RR (PMID: 33519504, PMID: 33071812)
Response 1: As suggested, we proceed with the adjustment on p. 11, line 372: Another possibility to determine VT1 using HRV would be through the short-term scaling exponent alpha1 (DFA a1) [1]. However, it is still a recent proposal, in which the authors themselves suggest that the use of this index still needs to be better studied [1] since it appears to be influenced by spontaneous breathing during exercise [2], whereas this is less influenced when using RMSSD or Poincaré plot [3].
Specific Comments
Point 2) Abstract
- Line 12, please correct "34" to text.
- Please introduce CV and TE for T1 and T2 in the abstract
- Report the lack of differences in velocities for T1 and T2.
Response 2:
As suggested, we proceed with the adjustment:
- a) Line 12: thirty-four males.
- b) T1 (CV > 12 % and TE >10%), T2 (CV < 12 % and TE < 10%)
- c) No difference was found to speed and V̇O2 for T1 and T2.
Point 3) Introduction
- Lines 52-54, Some recent papers indicate that DFA analysis over RR interval is a useful tool to monitor LT1 and VT1. Why did the authors not implement that analysis for HRV1 (i.e., HR complexity analysis)
Response 3:
We are grateful for the suggestion of DFA a1. However, our study was based on studies from 1998 to 2017 that compared thresholds [4–12]. These studies have commented on, used, or even validated, in some, the use of the time domain or Poincaré plot [4–12]. Furthermore, this demonstrates that the DFA a1 does not invalidate the indices used by our study; on the contrary, it is only a new possibility. In addition, research shows that the frequency domain has been observed at times as an unreliable value [13]. In contrast, they state that time-domain measurements are closely related to VT1 but have little verification [1,8], thus necessitating more research using this domain. Furthermore, work also shows that Poincaré's Plot (attributed to the non-linear index) has potential as a marker of low intensity [14], although it is not considered a good index to measure in young athletes with an approximate age of 15.43 ± 0.51 years [15]. Therefore, our study was based on adult, non-athlete individuals with a mean age of 22 ± 2 years, and we chose to use the time domain as they had only a few studies using this way combined with confirmation by the Poincaré Plot.
Point 4) Materials and Methods
- Was the study approved by an ethical committee? Please indicate it in the text.
- Line 98-99, please report ICC between evaluators for each of the six variables separately.
- Was the velocity/power on the treadmill registered?
- Anthropometry, Lines 105-108, the formulae (and reference) employed to calculate the body fat should be reported.
- Lines 132-133, Please report the participants' initial velocity (mean and SD) employed.
- HRVTs determination, Lines 178-182, How was the linearity breakpoint determined by visual inspection or some mathematical function? Please clarify that point.
Response 4:
As suggested, we proceed with the adjustment:
- a) Inserted on p. 2, line 83: and approved by an ethical committee (CAAE: 76607717.5.0000.5542).
- b) (LT1: 0.889, LT2: 0.948; VT1: 0.965, VT2: 0.954; HRVT1: 1.00, HRVT2: 1.00).
- c) Inserted on p. 3, line 126: and speed
- d) To calculate %BF, it is necessary to calculate body density (BD), Jackson & Pollock's seven skinfold equation was used: BD = 1.11200000 – 0.00043499 (X1) + 0.00000055 (X1)2 - 0.00028826 (X4), where X1 = sum of seven skinfolds and X4 = age in Years [16]. The %BF for the anthropometric technique was estimated using the SIRI (1961) equation %BF = (495/BD) - 450 [17].
- e) The average initial velocity of participants in the CPxS was 5.3 ± 1.2 km·h-1.
- f) HRVT2 was determined by the linearity breakpoint after the lowest value with a subsequent increase in the RMSSD and confirmed by the linearity break of the SD1/SD2 variables just when necessary (determined by visual inspection).
Point 5) Results
-In table 2 and 3, please explain % variables (Speed, VO2, HR), the reviewer guesses the percentage is relative to the maximum value reached during the same test, but it should be stated to avoid confusion.
Response 5:
As suggested, we proceed with the adjustment: Inserted on p. 6, line 241: The percentage demonstrated in Tables 2 and 3 (speed, V̇O2, RR, and HR) is relative to the maximum value reached during the CPxS.
Point 6) References
- Lines 471, please correct reference 7, the title is missing
Response 6:
As suggested, we proceed with the adjustment: Caen, K.; Vermeire, K.; Bourgois, J.G.; Boone, J. Exercise Thresholds on Trial: Are They Really Equivalent? Med Sci Sports Exerc 2018, 50, 1277–1284, doi:10.1249/MSS.0000000000001547.
References used in this file:
- Rogers, B.; Giles, D.; Draper, N.; Hoos, O.; Gronwald, T. A New Detection Method Defining the Aerobic Threshold for Endurance Exercise and Training Prescription Based on Fractal Correlation Properties of Heart Rate Variability. Front Physiol 2021, 11, 1806, doi:10.3389/FPHYS.2020.596567/XML/NLM.
- Gronwald, T.; Rogers, B.; Hoos, O. Fractal Correlation Properties of Heart Rate Variability: A New Biomarker for Intensity Distribution in Endurance Exercise and Training Prescription? Front Physiol 2020, 11, doi:10.3389/fphys.2020.550572.
- Penttilä, J.; Helminen, A.; Jartti, T.; Kuusela, T.; Huikuri, H. v.; Tulppo, M.P.; Coffeng, R.; Scheinin, H. Time Domain, Geometrical and Frequency Domain Analysis of Cardiac Vagal Outflow: Effects of Various Respiratory Patterns. Clinical Physiology 2001, 21, 365–376, doi:10.1046/j.1365-2281.2001.00337.x.
- Leprêtre, P.-M.; Bulvestre, M.; Ghannem, M.; Ahmaidi, S.; Weissland, T.; Lopes, P. Determination of Ventilatory Threshold Using Heart Rate Variability in Patients with Heart Failure. Surgery:Current Research 2013, 01, 1–6, doi:10.4172/2161-1076.S12-003.
- Nascimento, E.M.F.; Kiss, M.A.P.D.M.; Santos, T.M.; Lambert, M.; Pires, F.O. Determination of Lactate Thresholds in Maximal Running Test by Heart Rate Variability Data Set. Asian J Sports Med 2017, 8, doi:10.5812/asjsm.58480.
- Candido, N.; Okuno, N.; da Silva, C.; Machado, F.; Nakamura, F. Reliability of the Heart Rate Variability Threshold Using Visual Inspection and Dmax Methods. Int J Sports Med 2015, 36, 1076–1080, doi:10.1055/s-0035-1554642.
- Mankowski, R.T.; Michael, S.; Rozenberg, R.; Stokla, S.; Stam, H.J.; Praet, S.F.E. Heart-Rate Variability Threshold as an Alternative for Spiro-Ergometry Testing: A Validation Study. J Strength Cond Res 2016, 1, doi:10.1519/JSC.0000000000001502.
- Karapetian, G.K.; Engels, H.J.; Gretebeck, R.J. Use of Heart Rate Variability to Estimate LT and VT. Int J Sports Med 2008, 29, 652–657, doi:10.1055/s-2007-989423.
- Tulppo, M.P.; Mäkikallio, T.H.; Seppänen, T.; Laukkanen, R.T.; Huikuri, H. v; Coote, J.H.; Fisher, J.P.; Ogoh, S.; Ahmed, A.; Aro, M.R.; et al. Vagal Modulation of Heart Rate during Exercise: Effects of Age and Physical Fitness. The American Physiological Society 1998, 424–429.
- Mourot, L.; Fabre, N.; Savoldelli, A.; Schena, F. Second Ventilatory Threshold from Heart-Rate Variability: Valid When the Upper Body Is Involved? Int J Sports Physiol Perform 2014, 9, 695–701, doi:10.1123/IJSPP.2013-0286.
- Ramos-Campo, D.J.; Rubio-Arias, J.A.; Ávila-Gandía, V.; Marín-Pagán, C.; Luque, A.; Alcaraz, P.E. Heart Rate Variability to Assess Ventilatory Thresholds in Professional Basketball Players. J Sport Health Sci 2017, 6, 468–473, doi:10.1016/j.jshs.2016.01.002.
- Mendia-Iztueta, I.; Monahan, K.; Kyröläinen, H.; Hynynen, E. Assessment of Heart Rate Variability Thresholds from Incremental Treadmill Tests in Five Cross-Country Skiing Techniques. PLoS One 2016, 11, 1–14, doi:10.1371/journal.pone.0145875.
- Cottin, F.; Médigue, C.; Lopes, P.; Leprêtre, P.M.; Heubert, R.; Billat, V. Ventilatory Thresholds Assessment from Heart Rate Variability during an Incremental Exhaustive Running Test. Int J Sports Med 2007, 28, 287–294, doi:10.1055/s-2006-924355.
- Tulppo, M.P.; Makikallio, T.H. Quantitative Beat-to-Beat Analysis of Heart Rate Dynamics during Exercise. The American Physiological Society 1996, 244–252.
- Blasco-Lafarga, C.; Camarena, B.; Mateo-March, M. Cardiovascular and Autonomic Responses to a Maximal Exercise Test in Elite Youngsters. Int J Sports Med 2017, 38, 666–674, doi:10.1055/S-0043-110680/ID/R5999-0026.
- Jackson, A.S.; Pollock, M.L. Practical Assessment of Body Composition. Phys Sportsmed 1985, 13, 76–90, doi:10.1080/00913847.1985.11708790.
- Siri, W.E. Body Composition from Fluid Spaces and Density: Analysis of Methods. Nutrition 1961, 9, 480–491; discussion 480, 492.
Reviewer 2 Report
1) The authors should mention the precise heart rate variability and lactate thresholds used in the study to delineate between low-intensity and high-intensity exercise;
2) “No differences were found in threshold 2 (T2) compared to all methods” – this statement should be revised, as it is not clear which thresholds were similar (HRV, lactate). Also, it is not clear what “all methods” means;
3) The novelty of the study should be highlighted in the introduction section, as compared to other published articles (eg., https://doi.org/10.3389/fphys.2020.596567);
4) The inclusion criteria should be more detailed: previous physical condition of the subjects, consume of alcohol, coffee or other stimulants before testing which could affect HRV measurements;
5) The authors should mention methods used to collect HRV data from participants and what parameters were investigated (time-domain, frequency-domain, or non-linear HRV parameters);
6) It would be interesting to discuss practical clinical implications of their findings, in particular that HRVT2 was a valid and noninvasive method that can estimate LT2 and VT2;
7) It would be useful an authors’ opinion on HRV assessment in patients with cardiovascular diseases scheduled for cardiopulmonary exercise testing as compared to healthy individuals.
Author Response
We would like to thank the reviewer for evaluating our manuscript. We have addressed all the reviewers' concerns.
Response to Reviewer 2 Comments
Point 1) The authors should mention the precise heart rate variability and lactate thresholds used in the study to delineate between low-intensity and high-intensity exercise;
Response 1:
We do not understand the question, but if you refer to intensity equivalence moments with the literature, it follows: The intensity at T1 (LT1, VT1, HRVT1) was between 51-56% of V̇O2max, and T2 (LT2, VT2, HRVT2) was between 82-84% of V̇O2max. These findings agree with the literature that defines T1 between 50-60% and T2 above 80% of V̇O2max [1,2]
Point 2) “No differences were found in threshold 2 (T2) compared to all methods” – this statement should be revised, as it is not clear which thresholds were similar (HRV, lactate). Also, it is not clear what “all methods” means;
Response 2:
As suggested, we proceed with the adjustment on p. 1, lines 19-20: No differences were found in threshold 2 (T2) between LT2, VT2, and HRVT2.
Point 3) The novelty of the study should be highlighted in the introduction section, as compared to other published articles (eg., https://doi.org/10.3389/fphys.2020.596567);
Response 3:
As suggested, we proceed with the adjustment: p. 2, line 69: Therefore, it is interesting to present and compare the six thresholds in the same individual, allowing a more significant contribution to studies that demonstrate the thresholds in isolation, such as the comparison only between the first thresholds or only the second ones, and even only the comparison of HRV with the ventilatory threshold or with lactate threshold alone.
Point 4) The inclusion criteria should be more detailed: previous physical condition of the subjects, consume of alcohol, coffee or other stimulants before testing which could affect HRV measurements;
Response 4:
As suggested, we proceed with the adjustment on p. 2, lines 88: The study consisted of 34 healthy male university students physically independent (≥150 min·week-1 of physical exercise),
p.2, lines 93-94: Alcohol was forbidden for 48 h, and caffeine containing products for 24 h prior to the beginning of the study.
Point 5) The authors should mention methods used to collect HRV data from participants and what parameters were investigated (time-domain, frequency-domain, or non-linear HRV parameters);
Response 5:
The paper already mentions about methods: p. 3, lines 146-151: “A Polar H7 Bluetooth heart rate monitor (Polar Electro Oy, Kempele, Finland) was connected to a smartphone to collect HRV data, beat by beat (R-R intervals). The Polar H7 was validated to measure HRV [3]. Each step described in CPxS was recorded in the Elite HRV app (HRV Elite, Asheville-North Carolina) [4] and edited in the Kubios HRV 3.0 Program. Data were collected at rest, during, and after CPxS.”
Moreover, on p. 4, lines 171-186: “The R-R intervals were grouped into three-minute sequences for HRV analysis. Data filtering was performed using the Kubios HRV Standard 3.0 Program (Gronau, Germany), and it was automatically filtered to remove missing or premature intervals and artifacts not exceeding 5% [5]. The first 90 seconds of physical effort at each stage were excluded from the analysis due to the HR and HRV kinetic adjustment. This study chose the Root Mean Square of the Successive Differences (RMSSD) and Poincaré plot indexes (SD1 and SD2) as the HRV indexes (time-domain and non-linear HRV parameters). HRVT1 was identified by the first linearity breakpoint determined by visual inspection for the RMSSD variables, and when necessary, SD1 was used to confirm the identification [6,7]. HRVT2 was determined by the linearity breakpoint after the lowest value with a subsequent increase in the RMSSD and confirmed by the linearity break of the SD1/SD2 variables just when necessary (determined by visual inspection) [7–9]. They were identified in Excel software (Microsoft Excel® 2022) (Figure 1). Only two evaluators were necessary for HRVTs identification because there was no disagreement. The same evaluators were used to determine VTs, LTs, and HRVTs, and they did not have experience with HRVTs. All thresholds (LT1/2, VT1/2, HRVT1/2) were represented in Figure 1 (individual example) in CPxS.”
-As suggested, we proceed with the adjustment: Inserted on p.4, line 178: (time-domain and non-linear HRV parameters).
Point 6) It would be interesting to discuss practical clinical implications of their findings, in particular that HRVT2 was a valid and noninvasive method that can estimate LT2 and VT2;
Response 6:
Thankful for your suggestion, we proceed with the adjustment on p. 12, line 421: “Practical Significance: These findings may be helpful to coaches, conditioning instructors, and laboratories' daily routines to monitor and prescribe exercise intensity. The HRV presented itself as a simple analysis and interpretation method, visually using the time domain and Poincaré plot. Furthermore, HRV measurement using the heart rate sensor via smartphone reduces the cost and facilitates its analysis, allowing less complexity to assess and help identify different intensity domains, which is essential for monitoring exercise intensity and prescriptions.”
Point 7) It would be useful an authors’ opinion on HRV assessment in patients with cardiovascular diseases scheduled for cardiopulmonary exercise testing as compared to healthy individuals.
Response 7:
Thankful for your suggestion, we proceed with the adjustment on p. 12, line 420: After all, T1 is an important index when it comes to specific populations, for example, in patients with cardiovascular disease (ischemia or heart failure), who require adequate workload control of low-intensity exercises (below T1) [10]. In addition, the position statement of the European Association for Cardiovascular Prevention and Rehabilitation suggests the use of threshold-based exercise intensity prescriptions in heart disease patients [11,12].
References used in this file:
- Jones, A.M.; Carter, H. The Effect of Endurance Training on Parameters of Aerobic Fitness. / Effet de l’entrainement d’endurance Sur Les Parametres de La Capacite Aerobie. Sports Medicine 2000, 29, 373–386.
- Poole, D.C.; Rossiter, H.B.; Brooks, G.A.; Gladden, L.B. The Anaerobic Threshold: 50+ Years of Controversy. Journal of Physiology 2021, 599, 737–767, doi:10.1113/JP279963.
- Plews, D.J.; Scott, B.; Altini, M.; Wood, M.; Kilding, A.E.; Laursen, P.B. Comparison of Heart-Rate-Variability Recording with Smartphone Photoplethysmography, Polar H7 Chest Strap, and Electrocardiography. Int J Sports Physiol Perform 2017, 12, 1324–1328, doi:10.1123/ijspp.2016-0668.
- Perrotta, A.S.; Jeklin, A.T.; Hives, B.A.; Meanwell, L.E.; Warburton, D.E.R. Validity of the Elite HRV Smartphone Application for Examining Heart Rate Variability in a Field-Based Setting. J Strength Cond Res 2017, 31, 2296–2302, doi:10.1519/JSC.0000000000001841.
- Karapetian, G.K.; Engels, H.J.; Gretebeck, R.J. Use of Heart Rate Variability to Estimate LT and VT. Int J Sports Med 2008, 29, 652–657, doi:10.1055/s-2007-989423.
- Leprêtre, P.-M.; Bulvestre, M.; Ghannem, M.; Ahmaidi, S.; Weissland, T.; Lopes, P. Determination of Ventilatory Threshold Using Heart Rate Variability in Patients with Heart Failure. Surgery:Current Research 2013, 01, 1–6, doi:10.4172/2161-1076.S12-003.
- Nascimento, E.M.F.; Kiss, M.A.P.D.M.; Santos, T.M.; Lambert, M.; Pires, F.O. Determination of Lactate Thresholds in Maximal Running Test by Heart Rate Variability Data Set. Asian J Sports Med 2017, 8, doi:10.5812/asjsm.58480.
- Candido, N.; Okuno, N.; da Silva, C.; Machado, F.; Nakamura, F. Reliability of the Heart Rate Variability Threshold Using Visual Inspection and Dmax Methods. Int J Sports Med 2015, 36, 1076–1080, doi:10.1055/s-0035-1554642.
- Mankowski, R.T.; Michael, S.; Rozenberg, R.; Stokla, S.; Stam, H.J.; Praet, S.F.E. Heart-Rate Variability Threshold as an Alternative for Spiro-Ergometry Testing: A Validation Study. J Strength Cond Res 2016, 1, doi:10.1519/JSC.0000000000001502.
- Rogers, B.; Mourot, L.; Gronwald, T. Aerobic Threshold Identification in a Cardiac Disease Population Based on Correlation Properties of Heart Rate Variability. Journal of Clinical Medicine 2021, Vol. 10, Page 4075 2021, 10, 4075, doi:10.3390/JCM10184075.
- Mezzani, A.; Hamm, L.F.; Jones, A.M.; McBride, P.E.; Moholdt, T.; Stone, J.A.; Urhausen, A.; Williams, M.A. Aerobic Exercise Intensity Assessment and Prescription in Cardiac Rehabilitation: A Joint Position Statement of the European Association for Cardiovascular Prevention and Rehabilitation, the American Association of Cardiovascular and Pulmonary Rehabilitation and the Canadian Association of Cardiac Rehabilitation. Eur J Prev Cardiol 2013, 20, 442–467, doi:10.1177/2047487312460484.
- Marcin, T.; Eser, P.; Prescott, E.; Prins, L.F.; Kolkman, E.; Bruins, W.; van der Velde, A.E.; Peña Gil, C.; Iliou, M.-C.; Ardissino, D.; et al. Training Intensity and Improvements in Exercise Capacity in Elderly Patients Undergoing European Cardiac Rehabilitation – the EU-CaRE Multicenter Cohort Study. PLoS One 2020, 15, e0242503, doi:10.1371/journal.pone.0242503.
Round 2
Reviewer 2 Report
Congrats! No further questions.